# Potential mechanisms and drug prediction of Rheumatoid Arthritis and primary Sjögren's Syndrome: A public databases-based study

Li Wu[1,2], Qi Wang[1,3,4], Qi-chao Gao[1], Gao-xiang Shi[1,5], Jing Li[2], Fu-rong Fan[6], Jing Wu[6], Pei-Feng He[3,4]*, Qi Yu[3,4]*

**1** School of Basic Medical Sciences, Shanxi Medical University, Taiyuan, China, **2** Department of Anesthesiology, Shanxi Provincial People's Hospital (Fifth Hospital) of Shanxi Medical University, Taiyuan, China, **3** Shanxi Key Laboratory of Big Data for Clinical Decision Research, Taiyuan, China, **4** School of Management, Shanxi Medical University, Taiyuan, China, **5** Department of Anaesthesia, Shanxi Bethune Hospital, Shanxi Academy of Medical Sciences, Tongji Shanxi Hospital, Third Hospital of Shanxi Medical University, Taiyuan, China, **6** Academy of Medical Sciences, Shanxi Medical University, Taiyuan, China

* yuqi@sxmu.edu.cn (QY); hepeifeng2006@126.com (PFH)

## Abstract

Rheumatoid arthritis (RA) and primary Sjögren's syndrome (pSS) are the most common systemic autoimmune diseases, and they are increasingly being recognized as occurring in the same patient population. These two diseases share several clinical features and laboratory parameters, but the exact mechanism of their co-pathogenesis remains unclear. The intention of this study was to investigate the common molecular mechanisms involved in RA and pSS using integrated bioinformatic analysis. RNA-seq data for RA and pSS were picked up from the Gene Expression Omnibus (GEO) database. Co-expression genes linked with RA and pSS were recognized using weighted gene co-expression network analysis (WGCNA) and differentially expressed gene (DEG) analysis. Then, we screened two public disease–gene interaction databases (GeneCards and Comparative Toxicogenomics Database) for common targets associated with RA and pSS. The DGIdb database was used to predict therapeutic drugs for RA and pSS. The Human microRNA Disease Database (HMDD) was used to screen out the common microRNAs associated with RA and pSS. Finally, a common miRNA–gene network was created using Cytoscape. Four *hub* genes (*CXCL10*, *GZMA*, *ITGA4*, and *PSMB9*) were obtained from the intersection of common genes from WGCNA, differential gene analysis and public databases. Twenty-four drugs corresponding to hub gene targets were predicted in the DGIdb database. Among the 24 drugs, five drugs had already been reported for the treatment of RA and pSS. Other drugs, such as bortezomib, carfilzomib, oprozomib, cyclosporine and zidovudine, may be ideal drugs for the future treatment of RA patients with pSS. According to the miRNA–gene network, hsa-mir-21 may play a significant role in the mechanisms shared by RA and pSS. In conclusion, we identified commom targets as potential biomarkers in RA and pSS from publicly available databases and predicted potential drugs based on the targets. A new understanding of the molecular mechanisms associated with RA and pSS is provided according to the miRNA–gene network.

**Data Availability Statement:** The datasets generated and analyzed during the current study are available in the [GEO database] [http://www.ncbi.nlm.nih.gov/geo], [CTD database] [http://

ctdbase.org/] and [GeneCards database] [https://www.genecards.org/]. All data generated or analyzed during this study are included as supplementary information files.

**Funding:** Qi Yu and Pei-Feng He were supported by grants from the Shanxi Province key research and development program (No. 201903D311011; 201803D31067). The funders had a role in study conception and design.

**Competing interests:** The authors have declared that no competing interests exist.

## Introduction

Rheumatoid arthritis (RA) is a highly prevalent chronic autoimmune disease caused by a failure of the immune system to tolerate itself. Clinically, it consists of multi-joint, symmetrical, and aggressive inflammation of the small joints, constantly coupled with the involvement of extra-articular organs, which can lead to joint deformities and functional impairments [1]. Although RA is believed to be caused by immune disorder, neuroendocrine and microvascular dysregulation [2], its exact etiology and pathogenesis remain unknown.

Primary Sjögren's syndrome (pSS) is a rheumatic autoimmune disorder in which mononuclear cells infiltrate exocrine glands, causing their dysfunction and destruction [3]. The main clinical feature is desiccation of the mouth and eyes due to involvement of the salivary and lacrimal glands [4]. In secondary Sjögren's syndrome, Sjögren's syndrome occurs in conjunction with another systemic autoimmune disease, such as RA or systemic lupus erythematosus [5]. Despite its unknown etiology, pSS is a multifactorial disease that involves genetic predispositions and environmental factors [6].

RA patients have a high prevalence of pSS, estimated to be up to 55% according to previous reports [7]. Clinical features that overlap can make it difficult to differentiate between pathologies in RA and pSS. Some symptoms shared by these two diseases are arthralgia, positive rheumatoid factor, and elevated immunoglobulins [8]. This makes it common for patients suffering from pSS to be misdiagnosed, underdiagnosed, or diagnosed too late in their illness [9]. Despite their genomic similarity, RA and pSS differ phenotypically, making it essential to gain insight into the biomarkers and pathways related to them at the transcriptomic level [10]. However, most previous studies are related to secondary Sjögren's syndrome in RA, and few articles have reported on the pathogenesis of the co-occurrence of RA and pSS [11].

Weighted Gene Co-expression Network Analysis (WGCNA) is a computational biology method used to explore patterns of gene correlations among different samples. It can be used to cluster gene modules that exhibit highly coordinated changes in various biological processes and correlate these gene modules with clinical feature data [12, 13]. By analyzing the results, important core modules and molecules can be identified. The introduction of co-expression networks has facilitated a network-based gene selection method, which can be used in basic medical research to identify specific candidate biomarkers [14] or immunotherapy targets [15].

The main computational process of WGCNA involves raising correlation values between genes to a power, resulting in a topological overlap matrix network that better fits the characteristics of a scale-free network [16]. Its characteristic is the strengthening of strong correlations and weakening of weak correlations. In this study, the expression profile data of RA and pSS were analyzed using the WGCNA analysis method. After calculating and analyzing the correlation between gene modules and clinical feature data, the relationship between each gene module and clinical features was clarified. Combined with the analysis of differentially expressed genes, the biological functions and target genes of RA and pSS were identified.

## Methods

### Data sources

The GEO datasets were obtained from the NCBI NLM Gene Expression Omnibus (GEO) database (http://www.ncbi.nlm.nih.gov/geo), which contains gene expression data submitted by researchers worldwide. Human-tested specimens were included in the search for related gene expression data series using the keywords "rheumatoid arthritis" and "primary Sjögren's syndrome". Finally, we downloaded the data series numbered GSE55235, GSE55457, GSE1919, GSE110169, GSE84844, GSE23117, GSE40611, and GSE84844. Beyond that, the co-

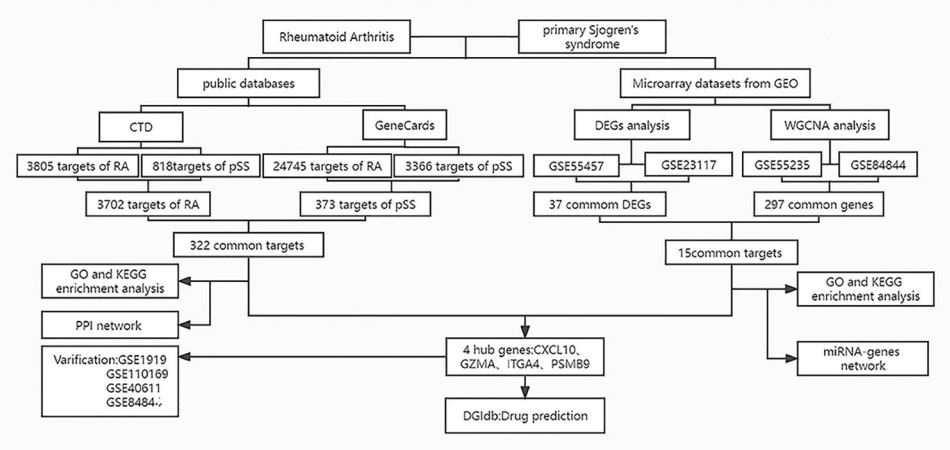

**Fig 1. Flow chart.**

expressed genes shared by RA and pSS were determined using the Comparative Toxicogenomics Database (CTD) (http://ctdbase.org/) [17] and the GeneCards Database (https://www.genecards.org/) [18]. The workflow of this study is shown in Fig 1.

## Weighted gene co-expression network analysis

The weighted gene co-expression network analysis (WGCNA) method was used to construct a network based on systematic gene expression levels to show co-expression relationships between genes. Analysis of gene expression patterns from multiple samples was conducted using this method, resulting in the clustering of genes with similar expression patterns. Phenotypic traits and gene association analyses were extensively performed using this method. The co-expression network was built using microarray data series GSE55235 and GSE84844 to obtain the RA and pSS-associated modules. We used the R package "WGCNA" to identify disease trait-related modules and hub genes [16]. First, the gene expression data was clustered to determine the optimal soft threshold (β) based on the average connectivity and scale-free topology fit index (R2). Multiple gene modules and hierarchical clustering trees were constructed. Then, Pearson correlation analysis was performed between genes to obtain a relationship matrix, and the module with the highest correlation was selected as the key gene module. Based on the criteria of a soft-thresholding power of 10 (scale free R2 = 0.85), a cut height of 0.25, and a minimum module size of 50, calculations were made to achieve scale-free topology for the GSE55235. For the GSE84844, we specified a soft-thresholding power of 4, a cut height of 0.4, and a minimum module size of 50 to confirm the co-expression modules. The genes selected for further analysis were those with strong correlation coefficients with clinical characteristics. We identified key modules in RA and pSS using Pearson correlation coefficients and the p-value of each eigengene and disease trait. Afterward, the genes in these key modules that were positively associated with RA and pSS were identified as shared genes.

## Differentially expressed genes analysis

We screened the differentially expressed genes (DEGs) in the GSE55457 and GSE23117 between the diseased and control groups in the expressing data using the "limma" R package [19]. The criteria for statistical significance were Log|FC| > 1 and adjusted p-value < 0.05. From a Venn diagram, we derived the common DEGs.

## Function enrichment analysis and protein–protein interaction network construction

To analyze the pathways and biological functions relevant to the respective common genes of the WGCNA and DEGs analysis, Gene Ontology (GO) and KEGG Enrichment Analysis (KEGG) [20] were conducted using the R package "clusterprofiler". An adjusted p < 0.05 in GO terms or KEGG pathways was considered statistically significant and visualized using the "GOplot" package under R 4.2.1version [21].

Using STRING (http://string-db.org), a tool for searching for interacting genes, a protein–protein interaction (PPI) network was developed. Using Cytoscape software (version 3.8.2), we constructed a PPI network with an interaction score greater than 0.7. Cluster analysis was conducted using a molecular complex detection plug-in (MCODE) with the following default parameters: K-core = 2, node score cutoff = 0.2, degree cutoff = 2, and maximum depth = 100.

## Targets from public databases

The common RA-related genes and pSS-related genes shared between the CTD database and the GeneCards database were obtained using Venn diagram. The CTD database integrated substantial data on the interaction between chemicals, genes, functional phenotypes, and diseases, which provided considerable superiorities for studying disease-related environmental exposure factors and potential mechanisms of drug action. The GeneCards database automatically integrated resources from approximately 150 gene-centric databases, including multi-faceted information on genomics, genetics, transcriptomics, proteomics, clinical, and functional aspects.

## Identification and validation of hub genes

By screening the CTD and GeneCards databases, shared targets between RA and pSS were identified. In addition, WGCNA and DEG analyses revealed shared genes. Then, hub genes were identified by combining shared targets from public disease databases with common genes identified by WGCNA and DEG analyses. Finally, four data series (GSE1919, GSE110169, GSE40611, and GSE84844) were examined to validate the hub genes.

## Drug prediction

The DGIdb database [22] (https://www.dgidb.org/) is a drug–gene interaction database that provides information about gene and drug interactions retrieved from databases, articles, and websites regarding interacting drugs by genes or interacting genes by drugs. With the DGIdb database, hub genes were used for potential candidate drugs for RA and pSS [23].

## Identification of common miRNAs

MiRNAs are a family of non-coding small RNAs consisting of approximately 22 nucleotides that regulate transcriptional processes after transcriptional initiation. Therefore, it was important to explore whether parts of miRNAs shared a similar developmental process and regulatory mechanism in RA and pSS. The Human microRNA Disease Database (HMDD) contains information regarding human miRNAs, miRNA disease-related data, evidence of functional abnormalities, and PubMed IDs [24]. The HMDD was used to search for miRNAs related to RA and pSS. The miRNAs were summarized, and the miRNAs that were confirmed by the literature to be related to having the same disorder types were selected as research objects.

**Table 1. Detailed information of GEO data series.**

| ID | GSE number | Platform | Samples | Disease |
|---|---|---|---|---|
| 1 | GSE55235 | GPL96 | 10 patients and 10 controls | Rheumatoid arthritis |
| 2 | GSE55457 | GPL96 | 13 patients and 10 controls | Rheumatoid arthritis |
| 3 | GSE1919 | GPL91 | 5 patients and 5 controls | Rheumatoid arthritis |
| 4 | GSE110169 | GPL13667 | 84 patients and 77 controls | Rheumatoid arthritis |
| 5 | GSE84844 | GPL570 | 30 patients and 30 controls | Primary Sjögren's syndrome |
| 6 | GSE23117 | GPL570 | 10 patients and 5 controls | Primary Sjögren's syndrome |
| 7 | GSE40611 | GPL570 | 17 patients and 18 controls | Primary Sjögren's syndrome |
| 8 | GSE84844 | GPL570 | 30 patients and 30 controls | Primary Sjögren's syndrome |

### Common miRNAs–genes network construction

The miRTarBase database (http://mirtarbase.mbc.nctu.edu.tw/php/index.php) was an informative resource for experimentally recognized miRNA–target interactions [25]. We collected information from the miRTarBase database regarding the target genes for common miRNAs. An miRNA–gene regulated network was created using common miRNA target genes in RA and pSS. The network was visualized using Cytoscape software.

## Results

### GEO information

A total of eight data series from the GEO database were downloaded: GSE55235, GSE55457, GSE1919, GSE110169, GSE84844, GSE23117, GSE40611and GSE84844. A detailed description of each of these data series could be found in Table 1, including GSE numbers, detection platforms, types of diseases, and samples. WGCNA analysis was conducted using GSE55235 and GSE84844. Meanwhile, DEG analysis was performed using GSE55457 and GSE40611, and the remaining data series were used to check the expression levels of the hub genes in the validation test.

### Co-expression network construction

By analyzing the GSE55235 with WGCNA, gene modules associated with RA were identified (Fig 2A–2D), including synovial tissue with 10 healthy joints and 10 RA joints. No poorly clustered samples were in the cluster analysis, so no model was deleted. A scale-free topology criterion was employed to figure the soft threshold. When soft threshold power $\beta = 10$ in the gene network, connectivity between genes was scale free (Fig 2A). Hierarchical clustering was performed on the modules. Similar gene expression patterns were observed between modules on the same branch (Fig 2B). A total of nine co-expression modules were recognized after highly similar modules were merged (Fig 2C). The "MEturquoise" and "MEblue" modules were positively associated with RA (MEturquoise module: r = 0.99, p = 3e-16; MEblue module: r = 0.63, p = 0.003) (Fig 2D). There were 829 genes in the turquoise module and 291 genes in the blue module, for a total of 1120 genes in the two modules for subsequent analysis.

By analyzing the GSE84844 with WGCNA, gene modules associated with pSS were identified (Fig 2E–2H), including whole blood with 30 healthy controls and 30 pSS patients. Clustering analysis found no outliers in the WGCNA analysis. The calculation revealed that the soft threshold power $\beta = 4$ was highly correlated and suitable for constructing a co-expression network. (Fig 2E). A total of 10 co-expression modules were recognized after highly similar modules were merged (Fig 2F). Taking a minimum of 50 genes as standards, we merged similar genes into each gene model using the dynamic pruning tree method (Fig 2G). Among these 10

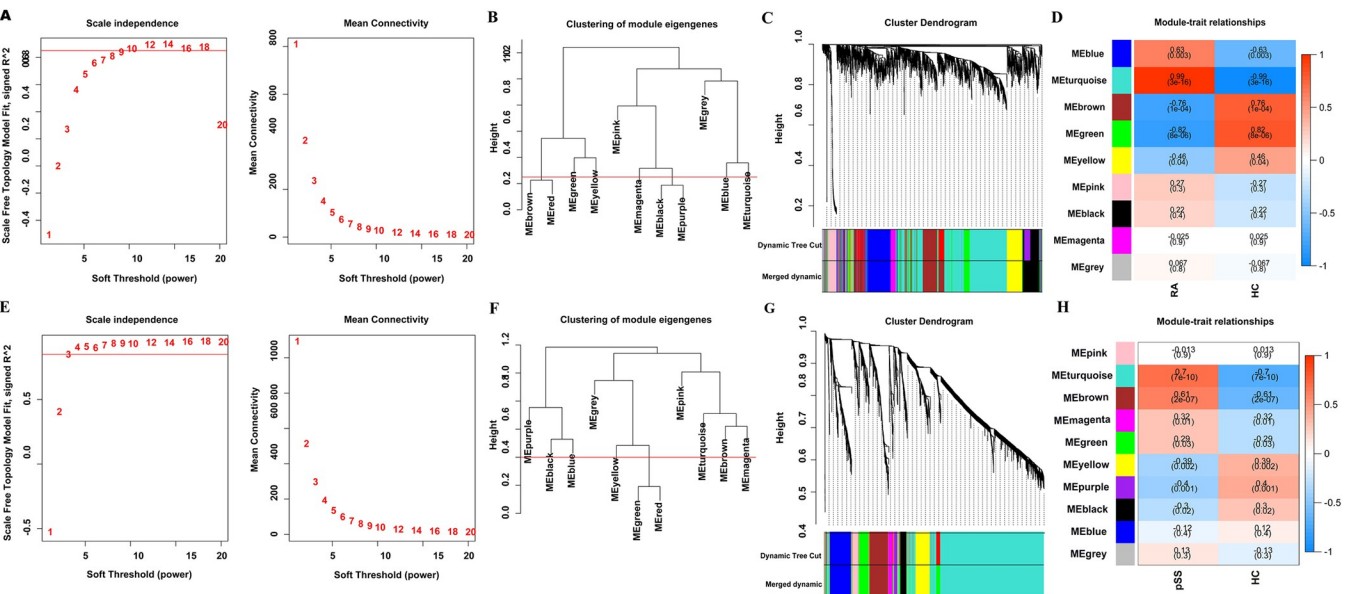

**Fig 2. WGCNA analysis of the GSE55235 and GSE84844 data series.** (A) Soft threshold analysis in RA. (B) Module correlations in RA. (C) Genes with differential expression measured in RA are clustered in the dendrogram. Color bands indicate the results obtained from automatic single-block analysis. (D) Heatmap of the module–trait relationship in RA. Each cell displays the corresponding correlation and p-value. (E) Soft threshold analysis in pSS. (F) Module correlations in pSS. (G) Genes with differential expression measured in pSS are clustered in the dendrogram. Color bands indicate the results obtained from automatic single-block analysis. (H) Heatmap of module–trait relationship in pSS. Each cell displays the corresponding correlation and p-value. RA, rheumatoid arthritis; pSS, primary Sjögren's syndrome; HC, healthy control.

modules, the "MEturquoise" and "MEbrown" modules had high correlations with pSS (MEturquoise module: r = 0.7, p = 7e-10; MEbrown module: r = 0.61, p = 2e-07). Using the dynamic pruning tree method, we merged genes that were similar in MEturquoise and MEbrown modules, including 2042 and 319 genes, respectively (Fig 2H).

## Enrichment analysis of common genes from WGCNA

The genes positively associated with the RA and pSS modules were identified as common genes. We obtained 297 common genes from the four positivity-related modules of RA and pSS (Fig 3A). The STRING online database was consulted to construct the PPI network based on these identified common genes. A visualization of the PPI network was derived from Cytoscape software (Fig 3B). As shown in Fig 3C, GO enrichment analysis indicated that response to virus, positive regulation of cytokine production, cellular response to cytokine stimulus, cell activation, and immune response-regulating signaling pathway were the primary functions of the common genes. KEGG enrichment analysis revealed that phagosomes, the p53 signaling pathway, and the MAPK signaling pathway were the primary functions (Fig 3D).

To explore the possible molecular mechanisms linked with RA and pSS, we performed a functional enrichment analysis of genes in their positive modules. The top 10 significantly enriched biological process terms (BPs), cellular components (CCs), and molecular functions (MFs) were shown in bar charts based on gene ontology enrichment analysis (S1 Fig in S1 File). GO enrichment analysis of the positive modules in RA showed that most genes were enriched within the immune inflammation pathways, such as positive regulation of cytokine production, leukocyte cell−cell adhesion, and regulation of T-cell activation. KEGG pathway enrichment results suggested that most target genes involved immune inflammation and signaling pathways, such as lipids, atherosclerosis, cytokine−cytokine receptor interaction, and

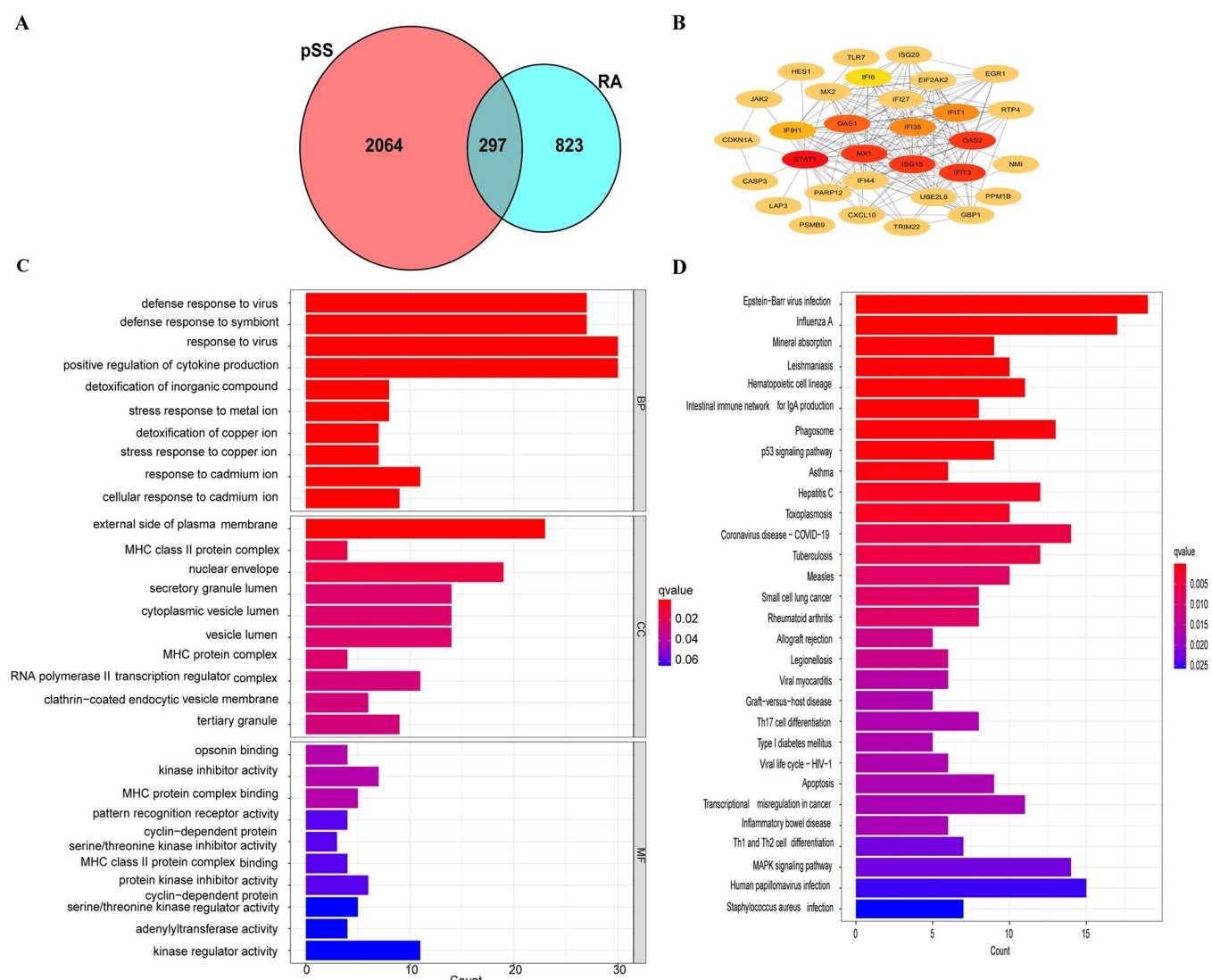

**Fig 3. Intersection genes of RA and pSS through WGCNA.** (A) Venn diagram of the intersection genes of RA and pSS. (B) PPI network of the intersection genes. (C) GO enrichment analysis of the intersection genes. (D) KEGG enrichment analysis of the intersection genes. RA, rheumatoid arthritis; pSS, primary Sjögren's syndrome.

the chemokine signaling pathway (S2 Fig in S1 File). KEGG pathway enrichment analyses in pSS revealed hub genes enriched for signaling pathways and RA (S3 Fig in S1 File). GO enrichment analysis revealed that these genes were primarily involved in response to virus, regulation of immune effector process, mitochondrial inner membrane, and transcription coregulator activity (S4 Fig in S1 File). The enrichment results for both diseases indicated that both were enriched in immune inflammatory pathways, and the enrichment results for pSS were directly enriched to RA. Therefore, RA and pSS might share a pathological mechanism, and immune and inflammatory responses might play a significant role.

## Identification of common DEGs

The GSE55457 and GSE23117 data series were used for DEGs analysis. The GSE55457 contained 13 RA and 10 normal controls from synovial membranes. The GSE23117 contained 10

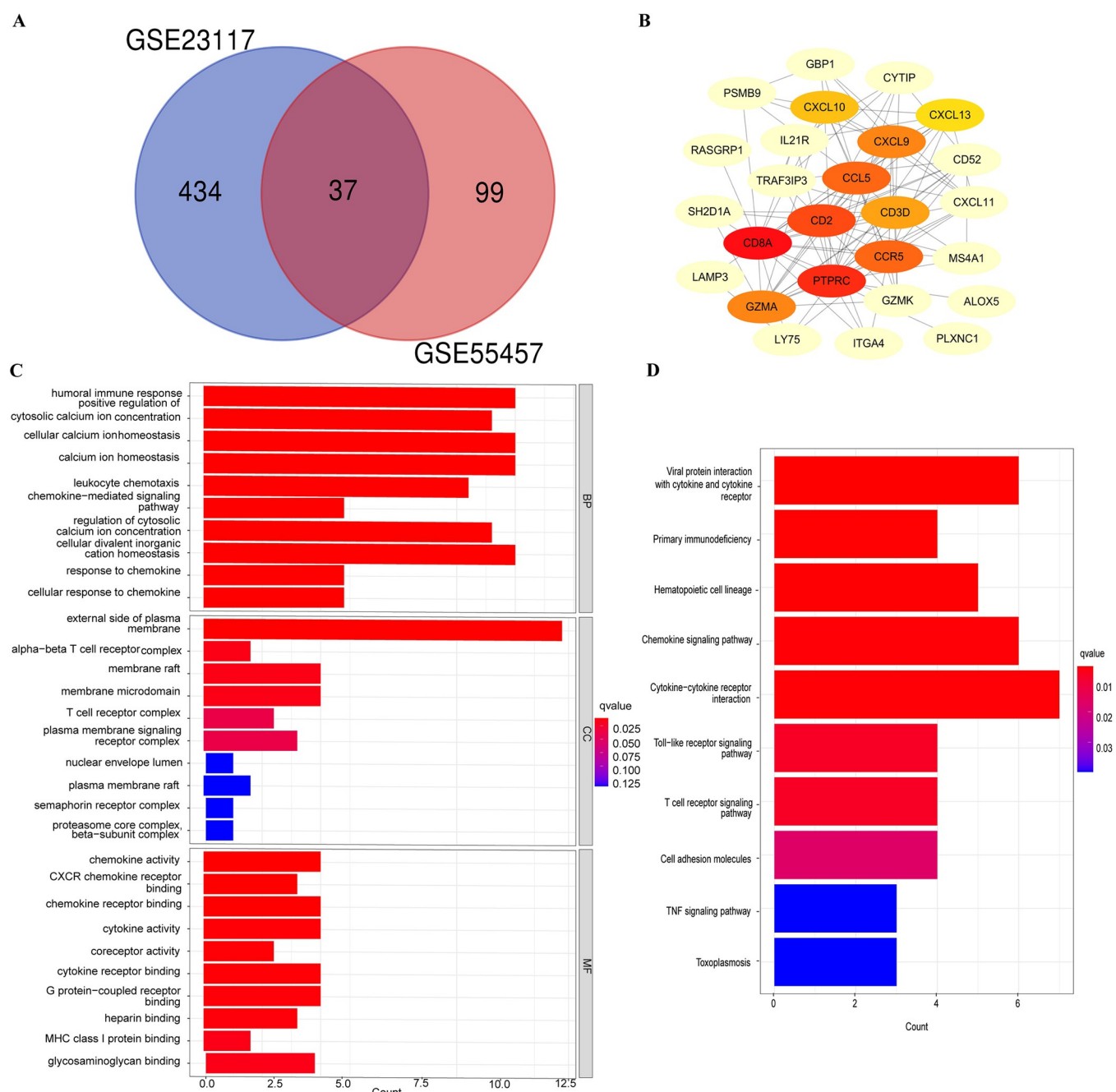

**Fig 4. Intersection DEGs between RA and pSS.** (A) Venn diagram of the intersection DEGs in GSE23117 and GSE55457. (B) PPI network of intersection DEGs. (C) GO enrichment analysis of the intersection DEGs. (D) KEGG enrichment analysis of the intersection DEGs.

pSS and 5 normal controls from the minor salivary glands. In order to identify DEGs, each set was normalized and analyzed with the limma R package. The GSE55457 and GSE23117 identified 137 and 472 DEGs, respectively. Intersection of the Venn diagrams revealed 37 DEGs that were commonly upregulated (Fig 4A). Then, based on intersection DEGs, the PPI network was built in Cytoscape 3.8.2, and the PPI network was imported (Fig 4B). According to the results of GO enrichment analysis, common differential genes were basically associated with

humoral immune response, cellular calcium ion homeostasis, and calcium ion homeostasis (Fig 4C). KEGG pathway enrichment analysis revealed common differential genes enriched for signaling pathways and primary immunodeficiency (Fig 4D). The results of enrichment analyses showed that immune and inflammation-related functions were again meaningfully enriched, in agreement with the WGCNA results.

## Gene targets from public databases

The CTD and GeneCards databases explored the interaction between potentially crucial genes in RA and pSS. Among the two databases, 3702 genes linked to RA and 372 genes linked to pSS were chosen. It was found that RA and pSS shared 322 gene targets (S1 Table in S1 File), which indicated that they shared a tremendous number of genes in common (Fig 5A). As a result of GO enrichment analysis, most of these genes were related to the immune and inflammation-related pathways, including positive regulation of cytokine production, leukocyte cell–cell adhesion, and the cytokine-mediated signaling pathway (Fig 5B). KEGG pathway enrichment analysis revealed genes enriched for signaling pathways, lipids, atherosclerosis, cytokine

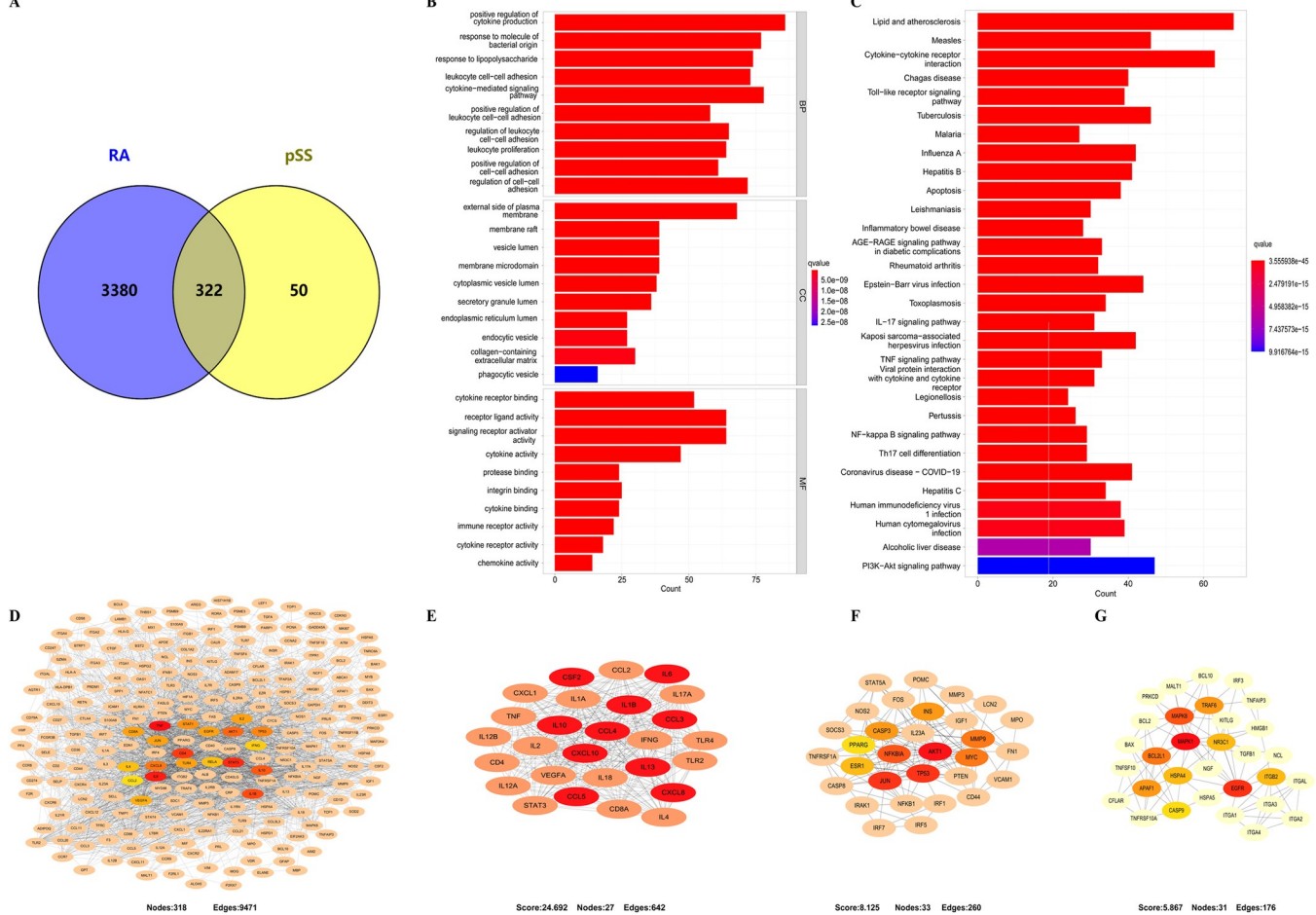

**Fig 5. Targets shared by RA and pSS from the CTD and GeneCards databases.** (A) Venn diagram of targets shared by RA and pSS from the CTD and GeneCards databases. (B) GO enrichment analysis of the shared targets. (C) KEGG enrichment analysis of the shared targets. (D) PPI network of the shared targets. (E–G) Three closely connected gene modules according to the MCODE plug-in for Cytoscape. RA, rheumatoid arthritis; pSS, primary Sjögren's syndrome.

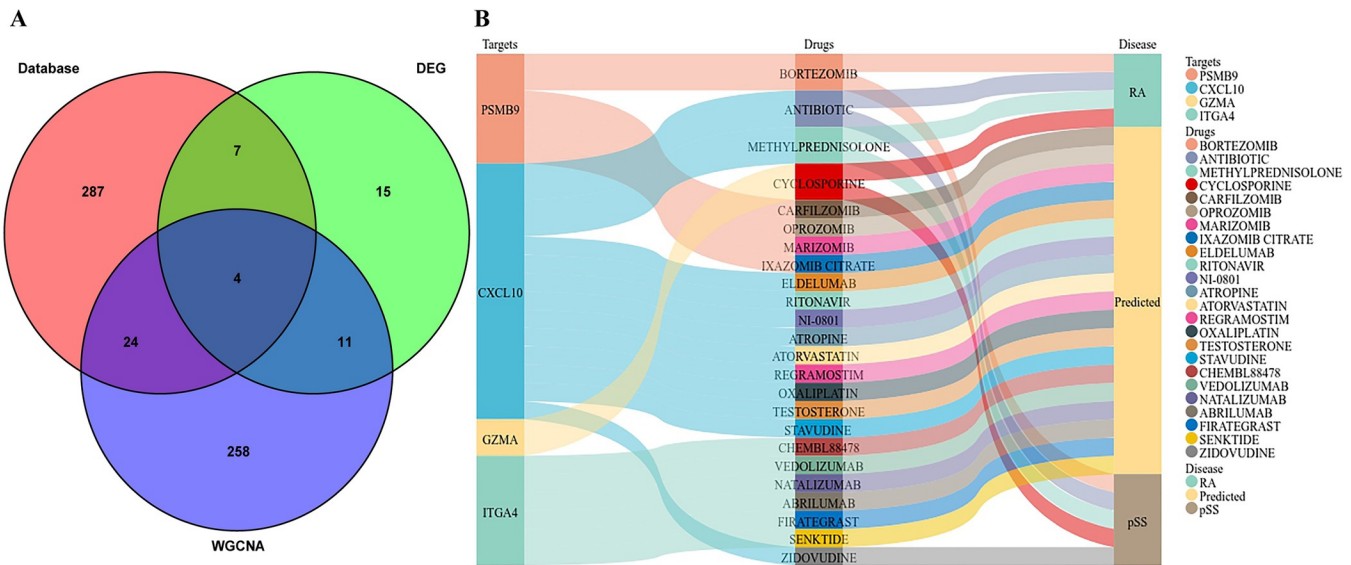

**Fig 6. Identification of hub genes.** (A) Venn diagram of targets shared in the CTD and GeneCards databases with genes from WGCNA and DEG analysis. (B) The Sankey diagram revealed the relationship between targets, drugs, and diseases. RA, rheumatoid arthritis; pSS, primary Sjögren's syndrome; WGCNA, weighted gene co-expression network analysis; DEG, differentially expressed genes.

−cytokine receptor interaction, etc. (Fig 5C). As with our enrichment analysis results, immune and inflammatory response-related functions were notably enhanced again.

Based on Cytoscape scores greater than 0.7, a PPI network was created for the common targets, containing 318 nodes and 9471 edges (Fig 5D). The MCODE plug-in for Cytoscape was employed to gain three closely connected gene modules (Fig 5E–5G). As a result of enrichment analysis, immune and inflammation-related functions were enriched again in these gene modules.

## Identification and validation of hub genes

The intersection of shared genes from WGCNA, differential genes, and public databases were used to obtain the hub genes. Eventually, four hub genes (*CXCL10*, *GZMA*, *ITGA4* and *PSMB9*) were obtained (Fig 6A). We analyzed the expression levels of the hub genes in RA using GSE1919 and GSE110169 to verify their reliability and the GSE40611 and GSE84844 in pSS. It was found in these data series that all four hub genes (*CXCL10*, *GZMA*, *ITGA4* and *PSMB9*) were up-regulated in both the RA and pSS groups compared with the control groups (Fig 7A–7D).

## Potential drugs for RA and pSS

Using the DGIdb database, we analyzed potential drugs for RA and pSS and obtained 24 predicted drugs corresponding to 4 hub genes (S2 Table in S1 File). The Sankey diagram revealed the relationships between drugs, diseases, and targets (Fig 6B). Among the predicted drugs, five drugs (bortezomib, antibiotic, methylprednisolone, cyclosporine and zidovudine) had already been documented or studied as targeted drugs for the treatment of RA and pSS, proving that the candidate drugs for RA and pSS had high credibility. There was a possibility that most potential drugs could interact with the hub gene, either in unknown ways or by inhibiting it.

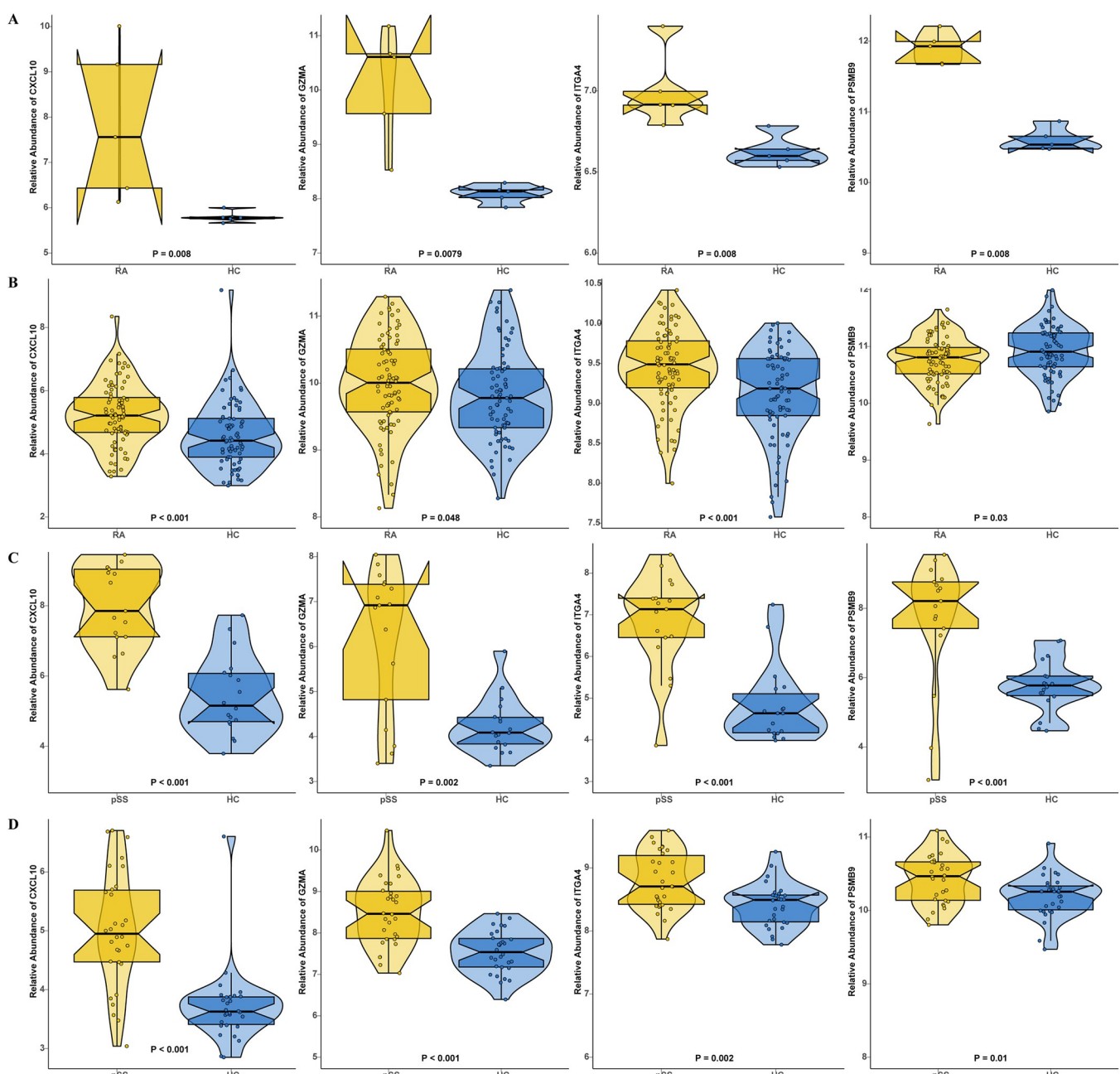

**Fig 7. Validation of hub genes.** (A) Hub gene expression in GSE1919. (B) Hub gene expression in GSE110169. (C) Hub gene expression in GSE40611. (D) Hub gene expression in GSE84844. Mean t-tests were used to compare the two sets of data, and statistical significance was determined by a p-value <0.05. RA, rheumatoid arthritis; pSS, primary Sjögren's syndrome.

## Identification of common miRNAs in RA and pSS

Based on the HMDD database, 149 miRNAs related to RA and 21 miRNAs related to pSS were identified. As a result of taking the intersection of miRNAs in RA and pSS, eight common miRNAs were identified. Based on literature mining, we got the types of miRNA disorders. There were six miRNAs (hsa-mir-126, hsa-mir-146a, hsa-mir-150, hsa-mir-155, hsa-mir-16

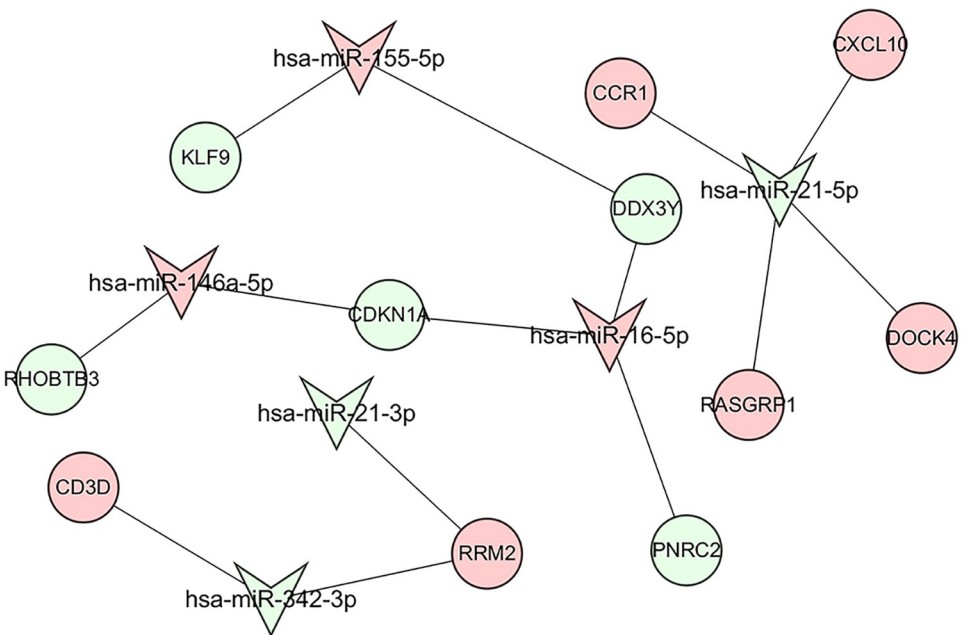

**Fig 8. MiRNA–gene regulatory network.** V shape indicates miRNA; circle shape indicates gene; green indicates downregulated; pink indicates upregulated.

and hsa-mir-223) upregulated and two miRNAs (hsa-mir-21 and hsa-mir-342) downregulated in both RA and pSS.

## Construction of the miRNA–hub genes network

From the miRTarbase database, we successfully identified 4341 target genes of eight common miRNAs. An miRNA–hub gene network was developed based on the intersection of these 4341 target genes and shared genes (generated by WGCNA and DEGs). Six miRNAs were in the miRNA–gene network, and 27 genes were shared among them, including one hub gene (*CXCL10*). Based on the network, it could be determined that hsa-miR-21-5p regulated the most downstream targets (Fig 8). Therefore, we speculated that hsa-mir-21 might contribute to the shared mechanisms of RA and pSS.

## Discussion

RA is an inflammatory disease that typically affects the joints, but it can also influence other tissues and organs. The incidence of sicca symptoms in patients with RA is relatively high [26, 27]. RA patients with pSS share the following clinical features: female predominance, high rheumatoid factor titers, high disease activity, articular erosions, and extra-articular manifestations [28–31]. RA patients with pSS are at an elevated risk of corneal injury, early tooth loss, oral infection, mucosa-associated lymphoid tissue (MALT) lymphoma, and mortality risk [32–34]. Therefore, it is essential to explore the relationship between RA and pSS in order to further understand their pathophysiology. Bioinformatics analysis based on computer algorithms of transcriptome data has been able to identify new biomarkers that have been applied to many immune-related diseases in recent years [35–37]. This is the first study to integrate data from multiple public databases to ascertain the shared mechanisms underlying RA and pSS.

In this study, we first completed WGCNA analysis using the RA and pSS-related microarray data sets in the GEO database to obtain valuable clues. A total of 297 genes significantly

related to RA and pSS were identified by the intersection of the modules. Based on GO analysis, these genes were primarily enriched in the cell cycle and immune inflammation signaling pathways, and KEGG enrichment analysis confirmed these results. In addition to WGCNA analysis, we conducted differential gene expression analysis of the two diseases, and 37 intersection genes were found. GO enrichment analysis determined that the most common differential genes were related to humoral immune response, cellular calcium ion homeostasis, and calcium ion homeostasis. Genes enriched for signaling pathways and primary immunodeficiency were identified by KEGG pathway enrichment analysis. Subsequently, based on our analysis of the relationship between WGCNA and DEGs, we identified 15 common genes. Then, using the CTD and GeneCards public databases, we explored the interaction between potentially crucial genes in RA and pSS. This research found that 322 genes are involved in both RA and pSS. Finally, we obtained the hub genes from the intersection of common targets from WGCNA, differential genes, and public databases. According to the results of this study, four hub genes (*CXCL10*, *GZMA*, *ITGA4* and *PSMB9*) could be used to determine disease activity in patients with RA and pSS, and these genes might provide insight into the common mechanisms underlying the two diseases.

The *CXCL10* protein has been categorized as a *Th1* chemokine that binds to the *CXCR3* receptor. It has been shown to control immune responses through activation and recruitment of leukocytes in RA and pSS patients [38]. There is evidence that *CXCL10* contributes to lymphoid homing and chronic inflammation persistence. Recent studies have shown that *CXCL10* is responsible for bone destruction through induction of the receptor activator of the nuclear factor kappa B (*NF-κB*) ligand in inflamed synovial tissue of RA patients [38]. Consistent with these findings, our study found that *CXCL10* expression was elevated in RA and pSS. *GZMA* is a serine protease secreted by cytotoxic lymphocytes that is involved in cell death, cytokine processing, and inflammation [39]. *GZMA* was shown to promote osteoclast precursor differentiation in mice by stimulating monocyte secretion of tumor necrosis factor alpha (*TNF-α*) and osteoclast precursors present in inflamed joints, leading to the development of RA [40]. Furthermore, granzyme A mRNA was found in lymphoid cells in the salivary glands of pSS patients but not in healthy people. This suggests that *GZMA* is at least one of the mechanisms of salivary gland body destruction in pSS patients and provides additional markers for the pathogenesis of pSS. Our study also showed that *GZMA* expression was elevated in RA and pSS, suggesting that *GZMA* may promote the occurrence and development of RA and pSS. *ITGA4*, also termed *CD49d*, an alpha chain of the *α4β1* integrin heterodimer was associated with aggressive biomarkers [41]. Previously, a variant of *ITGA4* was considered a susceptibility locus of autoimmune diseases, especially RA [42]. Research has shown an association between progressive disease and higher ITGA4 expression [43]. We found that *ITGA4* was a common susceptibility gene for RA and pSS at the transcriptome level, suggesting that *ITGA4* might be a marker of poor prognosis in RA with pSS, but this needs to be verified by further experiments. An immunoproteasome subunit called *PSMB9*, or recombinant proteasome beta type 9, displaced constitutively expressed subunits under pathological inflammatory conditions [44]. Although there are very few reports about the role of *PSMB9* in the onset and progression of RA and pSS, *PSMB9* has recently been shown to be connected to a variety of immune-related diseases, such as dermatomyositis and systemic lupus erythematosus [45]. For the first time, we found that *PSMB9* had high diagnostic potential for both RA and pSS, although it was not a specific biomarker for either disease. *PSMB9* provided an opportunity to investigate its role in the pathophysiology of RA and pSS and its role in diagnosis and differential diagnosis.

Despite current treatment options for disease-modifying anti-rheumatic drugs, approximately 50% of RA and pSS patients remained resistant to the drugs or showed insufficient suppression of disease activity [46]. We analyzed potential drugs for RA and pSS, and 24 drugs

corresponding to hub gene targets were found in the DGIdb database. Five of the 24 predicted drugs (bortezomib, antibiotic, methylprednisolone, cyclosporine and zidovudine) had already been documented or studied as potential treatment options for RA and pSS. As a protease inhibitor, bortezomib is a first-line agent for the treatment of multiple myeloma. However, bortezomib also significantly improved general symptoms in pSS patients, particularly fatigue, reduced serum globulin levels, and serum viscosity, according to previous reports [47]. In both animal experiments and clinical trials, bortezomib ameliorated symptoms after the first cycle and continued to improve gradually in patients with refractory RA [48]. In addition, we also predicted protease inhibitors, such as carfilzomib, oprozomib, and marizomib, as candidate drugs for the treatment of RA and pSS. As an antibiotic, minocycline is a semisynthetic derivative of tetracycline, and tetracyclines have been found to have anti-inflammatory and immunomodulatory properties. Tetracyclines are believed to inhibit matrix metalloproteinases, which might explain their effect in RA [49]. Previous studies have demonstrated the helpful effects of minocycline on almost all overall laboratory indexes and clinical symptoms in patients with RA when used as a single second-line agent or combined with other antirheumatic drugs [50]. Methylprednisolone is a glucocorticoid that can play anti-inflammatory and immunomodulatory roles in treating rheumatic diseases. It could effectively help patients relieve intermittent pain caused by rheumatic diseases. Cyclosporine is a traditional immunosuppressant, initially used in organ transplantation, that is now widely used in treating various autoimmune diseases, such as RA and pSS [51]. The 2020 European League Against Rheumatism recommendations for the management of pSS by topical and systemic therapy stated that cyclosporine showed acceptable safety as a drug that could be used topically [52]. Zidovudine and stavudine, thymidine analogues, are antiretroviral drugs that inhibit retroviral replication by disrupting viral reverse transcriptase and extending viral DNA strands [53]. A previous clinical trial yielded statistically significant results suggesting that zidovudine could be a feasible, effective, and low-risk treatment for pSS [54]. Other predictive drugs included several biological agents, such as vedolizumab, natalizumab, abrilumab, and eldelumab, which might help guide future trials.

Our study also demonstrated that hsa-miR-21 was involved in immune and inflammatory responses when we constructed miRNA–gene networks. Studies have found that abnormal expression of miR-21 was associated with a variety of cancers, including breast cancer [55], nasopharyngeal carcinoma [56], and renal clear cell carcinoma [57]. Recent studies have suggested that miR-21 is a particularly upregulated and active miRNA in psoriasis. By suppressing T cell apoptosis and targeting TIMP3, miR-21-5p promotes psoriatic skin inflammation [58]. However, the specific role of hsa-miR-21 in RA and pSS is not well understood. As a critical regulator of inflammation, hsa-miR-21 might be an essential target in the pathogenesis of RA and pSS. This provides us with new horizons for exploring the pathophysiological mechanisms of RA and pSS in the future.

The limitation of this study is that the severity of disease activity or clinical parameters of organ damage were not available from the publicly available databases, so we could not determine whether hub genes could be used as targets for the progression and invasion of diseases from the analysis. Therefore, more studies are needed to verify our results in cell and animal experiments, which will provide valuable guidance for future research.

## Conclusions

In conclusion, we comprehensively analyzed publicly available databases and gene expression microarray data from patients with RA or pSS and healthy controls. We identified four genes (*CXCL10*, *GZMA*, *ITGA4* and *PSMB9*) as potential biomarkers of disease activity in RA and

pSS. Potential drugs were predicted based on the common targets. A new understanding of the molecular mechanisms associated with RA and pSS is provided according to the miRNA–gene network.

## Supporting information

**S1 File.**
(DOCX)

## Author Contributions

**Conceptualization:** Qi Wang, Qi Yu.

**Data curation:** Li Wu, Qi-chao Gao, Gao-xiang Shi, Fu-rong Fan, Jing Wu.

**Formal analysis:** Li Wu, Qi-chao Gao, Gao-xiang Shi, Jing Li.

**Funding acquisition:** Pei-Feng He, Qi Yu.

**Investigation:** Qi-chao Gao.

**Methodology:** Li Wu, Jing Li.

**Project administration:** Pei-Feng He, Qi Yu.

**Resources:** Qi Yu.

**Supervision:** Qi Wang, Pei-Feng He, Qi Yu.

**Validation:** Li Wu.

**Writing – original draft:** Li Wu.

**Writing – review & editing:** Li Wu, Qi Wang.

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
