## [Decision Letter · Decision Letter 0]

25 Oct 2023

PONE-D-23-23956Potential mechanisms and drug prediction of Rheumatoid Arthritis and primary Sjögren’s Syndrome : a public databases-based studyPLOS ONE

Dear Dr. Yu,

Thank you for submitting your manuscript to PLOS ONE. After careful consideration, we feel that it has merit but does not fully meet PLOS ONE’s publication criteria as it currently stands. Therefore, we invite you to submit a revised version of the manuscript that addresses the points raised during the review process.

We look forward to receiving your revised manuscript.

Kind regards,

Gurudeeban Selvaraj

Academic Editor

PLOS ONE

Journal Requirements:

"This project was supported by grants from the Shanxi Province key research and development program (No. 201903D311011; 201803D31067)."

Reviewers' comments:

Reviewer's Responses to Questions

**Comments to the Author**

1. Is the manuscript technically sound, and do the data support the conclusions?

Reviewer #1: Yes

Reviewer #2: Yes

2. Has the statistical analysis been performed appropriately and rigorously? 

Reviewer #1: Yes

Reviewer #2: I Don't Know

3. Have the authors made all data underlying the findings in their manuscript fully available?

Reviewer #1: Yes

Reviewer #2: Yes

4. Is the manuscript presented in an intelligible fashion and written in standard English?

Reviewer #1: Yes

Reviewer #2: Yes

5. Review Comments to the Author

Reviewer #1: Regarding the manuscript entitled"

Potential mechanisms and drug prediction of Rheumatoid Arthritis and primary Sjögren’s Syndrome : a public databases-based study. The manuscript is interesting, but the quality of the figures should be improved.

Reviewer #2: Please provide the flowchart of study in a separate figure.

In the introduction section please provide brief description about WGCNA method. Reading the paper PMID: 37636389 is suggested.

" GSE55235, GSE55457, GSE1919, GSE110169, GSE84844, GSE23117, GSE40611, and GSE84844" are data series not data set please revise related sentences.

In the senetence "The co-expression network was built using microarray dataset GSE55235 and dataset GSE84844 to obtain the RA and pSS-associated modules" did the authors perform WGCNA on selected genes of data series . How did they screen genes for WGCNA?

In sentence "An adjusted p < 0.05 in GO terms or KEGG pathways was considered statistically significant and visualized using the “GOplot” package " please mention GOplot package under R software .

6. PLOS authors have the option to publish the peer review history of their article (what does this mean?). If published, this will include your full peer review and any attached files.

Reviewer #1: **Yes: **Shahram Teimourian

Reviewer #2: No

---

## [Author Response · Author response to Decision Letter 0]

31 Oct 2023

Dear editors and reviewers: 

Thank you very much for your careful review of our manuscript entitled “Potential mechanisms and drug prediction of Rheumatoid Arthritis and primary Sjögren’s Syndrome: a public databases-based study”. We have carefully considered the reviewers’ professional comments and made a point to point response as follows: 

To Reviewer 1： 

1. Regarding the manuscript entitled “Potential mechanisms and drug prediction of Rheumatoid Arthritis and primary Sjögren’s Syndrome : a public databases-based study” . The manuscript is interesting, but the quality of the figures should be improved. 

Response：Thank you for your professional suggestions. Later, I will re-upload high-resolution figures in the manuscript that meet the requirements of the magazine.

To Reviewer 2：

1. Please provide the flowchart of study in a separate figure.

Response：Thank you for your professional suggestions. Fig 1 in the manuscript is the flow chart of this study, and I will upload a separate flow chart again in the system.

2. In the introduction section please provide brief description about WGCNA method. Reading the paper PMID: 37636389 is suggested.

Response: Thank you for your professional suggestions. We describe the WGCNA approach in the introduction section. Weighted Gene Co-expression Network Analysis (WGCNA) is a computational biology method used to explore patterns of gene correlations among different samples. It can be used to cluster gene modules that exhibit highly coordinated changes in various biological processes and correlate these gene modules with clinical feature data [1, 2]. By analyzing the results, important core modules and molecules can be identified. The introduction of co-expression networks has facilitated a network-based gene selection method, which can be used in basic medical research to identify specific candidate biomarkers [3] or immunotherapy targets [4].

The main computational process of WGCNA involves raising correlation values between genes to a power, resulting in a topological overlap matrix network that better fits the characteristics of a scale-free network [5]. Its characteristic is the strengthening of strong correlations and weakening of weak correlations. In this study, the expression profile data of RA and pSS were analyzed using the WGCNA analysis method. After calculating and analyzing the correlation between gene modules and clinical feature data, the relationship between each gene module and clinical features was clarified. Combined with the analysis of differentially expressed genes, the biological functions and target genes of RA and pSS were identified.

References

1. Zhang T, Wong G. Gene expression data analysis using Hellinger correlation in weighted gene co-expression networks (WGCNA). Computational and structural biotechnology journal. 2022;20:3851-63.

2. Liu K, Chen S, Lu R. Identification of important genes related to ferroptosis and hypoxia in acute myocardial infarction based on WGCNA. Bioengineered. 2021;12(1):7950-63.

3. Ye H, Wang R, Wei J, Wang Y, Zhang X, Wang L. Bioinformatics Analysis Identifies Potential Ferroptosis Key Gene in Type 2 Diabetic Islet Dysfunction. Frontiers in endocrinology. 2022;13:904312.

4. Afshar S, Leili T, Amini P, Dinu I. Introducing novel key genes and transcription factors associated with rectal cancer response to chemoradiation through co-expression network analysis. Heliyon. 2023;9(8):e18869.

5. Langfelder P, Horvath S. WGCNA: an R package for weighted correlation network analysis. BMC Bioinformatics. 2008;9:559.

3. "GSE55235, GSE55457, GSE1919, GSE110169, GSE84844, GSE23117, GSE40611, and GSE84844" are data series not data set please revise related sentences.

Response：Thank you for your professional suggestions. Finally, we downloaded the data series numbered GSE55235, GSE55457, GSE1919, GSE110169, GSE84844, GSE23117, GSE40611, and GSE84844.

4. In the senetence "The co-expression network was built using microarray dataset GSE55235 and dataset GSE84844 to obtain the RA and pSS-associated modules" did the authors perform WGCNA on selected genes of data series . How did they screen genes for WGCNA?

Response：Thank you for your professional suggestions. We performed WGCNA on selected genes of data series GSE55235 and GSE84844. We used the R package “WGCNA” to identify disease trait-related modules and hub genes. First, the gene expression data was clustered to determine the optimal soft threshold (β) based on the average connectivity and scale-free topology fit index (R2). Multiple gene modules and hierarchical clustering trees were constructed. Then, Pearson correlation analysis was performed between genes to obtain a relationship matrix, and the module with the highest correlation was selected as the key gene module.

Based on the criteria of a soft-thresholding power of 10 (scale free R2 = 0.85), a cut height of 0.25, and a minimum module size of 50, calculations were made to achieve scale-free topology for the dataset GSE55235. For the dataset GSE84844, we specified a soft-thresholding power of 4, a cut height of 0.4, and a minimum module size of 50 to confirm the co-expression modules. The genes selected for further analysis were those with strong correlation coefficients with clinical characteristics. We identified key modules in RA and pSS using Pearson correlation coefficients and the p-value of each eigengene and disease trait. Afterward, the genes in these key modules that were positively associated with RA and pSS were identified as shared genes.

5. In sentence "An adjusted p < 0.05 in GO terms or KEGG pathways was considered statistically significant and visualized using the “GOplot” package " please mention GOplot package under R software.

Response：Thank you for your professional suggestions. An adjusted p < 0.05 in GO terms or KEGG pathways was considered statistically significant and visualized using the “GOplot” package under R 4.2.1 version.

---

## [Decision Letter · Decision Letter 1]

27 Dec 2023

PONE-D-23-23956R1Potential mechanisms and drug prediction of Rheumatoid Arthritis and primary Sjögren’s Syndrome : a public databases-based studyPLOS ONE

Dear Dr. Yu,

Thank you for submitting your manuscript to PLOS ONE. After careful consideration, we feel that it has merit but does not fully meet PLOS ONE’s publication criteria as it currently stands. Therefore, we invite you to submit a revised version of the manuscript that addresses the points raised during the review process.

We look forward to receiving your revised manuscript.

Kind regards,

Gurudeeban Selvaraj

Academic Editor

PLOS ONE

Journal Requirements:

Reviewers' comments:

Reviewer's Responses to Questions

**Comments to the Author**

1. If the authors have adequately addressed your comments raised in a previous round of review and you feel that this manuscript is now acceptable for publication, you may indicate that here to bypass the “Comments to the Author” section, enter your conflict of interest statement in the “Confidential to Editor” section, and submit your "Accept" recommendation.

Reviewer #3: All comments have been addressed

Reviewer #4: All comments have been addressed

2. Is the manuscript technically sound, and do the data support the conclusions?

Reviewer #3: Yes

Reviewer #4: Yes

3. Has the statistical analysis been performed appropriately and rigorously? 

Reviewer #3: Yes

Reviewer #4: Yes

4. Have the authors made all data underlying the findings in their manuscript fully available?

Reviewer #3: Yes

Reviewer #4: Yes

5. Is the manuscript presented in an intelligible fashion and written in standard English?

Reviewer #3: Yes

Reviewer #4: Yes

6. Review Comments to the Author

Reviewer #3: 1- This study reports on the findings of the original research. Researchers have used public databases and bioinformatics techniques to identify potential mechanisms and drug predictions for RA and pSS. This new study can contribute to our understanding of the structural molecular mechanisms of these diseases.

2- This study is based on standards and detailed analysis. The authors used rigorous methods to examine the data and find common genes, pathways, and therapeutic targets. These methods include weighted gene co-expression network analysis (WGCNA), differential gene expression analysis, and functional enrichment analysis.

3- The data presented in the article confirm the conclusion. By combining data from different studies, the authors discovered four hub genes (CXCL10, GZMA, ITGA4, and PSMB9). It was found that these genes are upregulated in both RA and pSS, which supports these findings.

Reviewer #4: The contents of the article and the topic are interesting, and the revision had considered of the first reviewer's recommendation. However, I suggest that improving the quality of the figures in the article would be better. They are obscure that I cannot see them clearly.

7. PLOS authors have the option to publish the peer review history of their article (what does this mean?). If published, this will include your full peer review and any attached files.

Reviewer #3: **Yes: **Masoud Ghorbani

Reviewer #4: No

---

## [Author Response · Author response to Decision Letter 1]

11 Jan 2024

Dear editors and reviewers:

Thank you very much for your careful review of our manuscript entitled “Potential mechanisms and drug prediction of Rheumatoid Arthritis and primary Sjögren’s Syndrome: a public databases-based study”. We have carefully considered the reviewers’ professional comments and made a point to point response as follows: 

To Reviewer 3： 

1. (1) This study reports on the findings of the original research. Researchers have used public databases and bioinformatics techniques to identify potential mechanisms and drug predictions for RA and pSS. This new study can contribute to our understanding of the structural molecular mechanisms of these diseases.

(2) This study is based on standards and detailed analysis. The authors used rigorous methods to examine the data and find common genes, pathways, and therapeutic targets. These methods include weighted gene co-expression network analysis (WGCNA), differential gene expression analysis, and functional enrichment analysis.

(3) The data presented in the article confirm the conclusion. By combining data from different studies, the authors discovered four hub genes (CXCL10, GZMA, ITGA4, and PSMB9). It was found that these genes are upregulated in both RA and pSS, which supports these findings.

Response： Thank you for your encouraging feedback on our research. We are grateful for your positive assessment of our work and appreciate your constructive suggestions for improvement. Your insights will undoubtedly help us to further refine and strengthen our study. We will carefully consider and address all of your comments in the revised manuscript. Thank you once again for your support and guidance.

To Reviewer 4：

1. The contents of the article and the topic are interesting, and the revision had considered of the first reviewer's recommendation. However, I suggest that improving the quality of the figures in the article would be better. They are obscure that I cannot see them clearly.

Response：We appreciate the feedback regarding the image quality issue, and we have re-uploaded clearer images. It is important to note that the merged version of figures seen during the peer review process may have undergone system compression, leading to a decrease in image quality. This could be the primary reason for the diminished image quality. We will promptly address this issue by providing the responsible editor with high-resolution images that meet the journal’s requirements. We appreciate your constructive criticism and assure you that we are committed to improving the quality of our manuscript in response to your feedback.

Thank you very much for your attention and time. Look forward to hearing from you.

---

## [Decision Letter · Decision Letter 2]

25 Jan 2024

Potential mechanisms and drug prediction of Rheumatoid Arthritis and primary Sjögren’s Syndrome : a public databases-based study

PONE-D-23-23956R2

Dear Dr. Yu,

We’re pleased to inform you that your manuscript has been judged scientifically suitable for publication and will be formally accepted for publication once it meets all outstanding technical requirements.

Kind regards,

Dr. Gurudeeban Selvaraj, Ph.D.

Academic Editor

PLOS ONE

Additional Editor Comments (optional):

Reviewers' comments:

Reviewer's Responses to Questions

**Comments to the Author**

1. If the authors have adequately addressed your comments raised in a previous round of review and you feel that this manuscript is now acceptable for publication, you may indicate that here to bypass the “Comments to the Author” section, enter your conflict of interest statement in the “Confidential to Editor” section, and submit your "Accept" recommendation.

Reviewer #4: All comments have been addressed

2. Is the manuscript technically sound, and do the data support the conclusions?

Reviewer #4: Yes

3. Has the statistical analysis been performed appropriately and rigorously? 

Reviewer #4: Yes

4. Have the authors made all data underlying the findings in their manuscript fully available?

Reviewer #4: Yes

5. Is the manuscript presented in an intelligible fashion and written in standard English?

Reviewer #4: Yes

6. Review Comments to the Author

Reviewer #4: The authors has revised the problems that I proposed in the previous recommendations. Thus I suggest to accept this article.

7. PLOS authors have the option to publish the peer review history of their article (what does this mean?). If published, this will include your full peer review and any attached files.

Reviewer #4: No

---

## [Editor Report · Acceptance letter]

6 Feb 2024

PONE-D-23-23956R2 

PLOS ONE

Dear Dr. Yu, 

I'm pleased to inform you that your manuscript has been deemed suitable for publication in PLOS ONE. Congratulations! Your manuscript is now being handed over to our production team.

Kind regards, 

on behalf of

Dr. Gurudeeban Selvaraj 

Academic Editor

PLOS ONE